# Learn2Assemble with Structured Representations and Search for Robotic Architectural Construction

**Niklas Funk, Georgia Chalvatzaki, Boris Belousov, and Jan Peters**
Department of Computer Science, Technical University of Darmstadt, Germany
{niklas,georgia,boris}@robot-learning.de, mail@jan-peters.net

**Abstract:** Autonomous robotic assembly requires a well-orchestrated sequence of high-level actions and smooth manipulation executions. Learning to assemble complex 3D structures remains a challenging problem that requires drawing connections between target designs and building blocks, and creating valid assembly sequences considering structural stability and feasibility. To address the combinatorial complexity of the assembly tasks, we propose a multi-head attention graph representation that can be trained with reinforcement learning (RL) to encode the spatial relations and provide meaningful assembly actions. Combining structured representations with model-free RL and Monte-Carlo planning allows agents to operate with various target shapes and building block types. We design a hierarchical control framework that learns to sequence the building blocks to construct arbitrary 3D designs and ensures their feasibility, as we plan the geometric execution with the robot-in-the-loop. We demonstrate the flexibility of the proposed structured representation and our algorithmic solution in a series of simulated 3D assembly tasks with robotic evaluation, which showcases our method's ability to learn to construct stable structures with a large number of building blocks. Code and videos are available at: https://sites.google.com/view/learn2assemble

**Keywords:** Structured representations, Autonomous assembly, Manipulation

## 1 Introduction

Construction and manufacturing are becoming increasingly automated in the last decades. However, there is an essential need for sustainable autonomous architectural assembly [1], where a game-changer would come with intelligent robot assembly abilities that optimally decide over plans, actions, execution, and efficiency [2]. In this work, our main focus is the combinatorial optimization problem of autonomously assembling complex structures with robotic manipulators *without an a priori* defined task plan and goal poses for the sequential picking-placing actions. For constructing abstract target designs, we must consider the combinatorics of the growing action space w.r.t. the number of available modules and the size of the structure. Therefore, an effective representation of the assembly problem is essential. Moreover, the geometric execution of the picking and placing actions by the robot imposes constraints to the assembly sequence, as those actions are subject to the kinematic feasibility in the robot's workspace. Eventually, the problem of assembling structures lies in the area of long-horizon manipulation tasks, where most methods in the literature consider a known task plan, and focus on fine manipulability and structural stability, or learn action sequences from demonstrations [3, 4, 5, 6, 7, 8].

In this paper, we propose a novel algorithmic solution for robotic assembly that combines high-level decision-making on the construction sequence with the geometric execution by the robot that should ensure feasibility and stability. We propose a graph-based representation that captures the relations between target shapes and available building blocks. Notably, we design a multi-head attention-based graph neural network (GNN) architecture with a purposefully induced inductive bias for encoding the structural representation of the assembly task. The GNN is trained through reinforcement learning (RL) to explore feasible actions, resulting in an expressive and flexible representation. When combined as prior to Monte Carlo Tree Search (MCTS), it extrapolates to out-of-distribution (OOD) assembly tasks, i.e., tasks with a higher number and different types of blocks and vari-

able target shapes. Ultimately, we provide a solution for the motion generation of the sequential picking-placing actions when having the robot-in-the-loop to ensure reachable and feasible actions, which do not disrupt the assembly process. The proposed *learn2assemble* algorithm provides a flexible, autonomous robotic assembly agent for constructing 3D shapes using more than ten blocks.

Our main contributions are threefold. (1) We propose a **multi-head attention-based graph representation** for the 3D assembly problem that is flexible enough for representing arbitrary, stable structures and their relations to different building blocks. (2) We design an **integrated long-horizon manipulation algorithm** that learns through **exploration**, which combined with **model-based search** leads to generalizable skills. Our method considers the **robot-in-the-loop**, integrating high-level action planning with low-level motion generation for learning policies that ensure the kinematic feasibility and stability of the constructed structure. Finally, (3) we developed a novel **benchmarking environment for 3D robot assembly**, which is modular for testing with and without the robot, for arbitrary target designs, and with adjustable types and number of objects. Our empirical results on a series of representative experiments showcase the generalization power of the proposed algorithmic solution, drawing interesting insights on the combination of integrated learning and planning for long-horizon manipulation that can apply to a range of robotic applications. **Related works.** Autonomous robotic assembly is essential for manufacturing and construction, and therefore, many works tried to tackle the problem of automatizing task execution and fine manipulation. Autonomous assembly is a long-horizon manipulation task with multiple stages of decisions and sub-task executions to be made alongside controlling the dynamic execution of the assembly process. Thus, researchers proposed

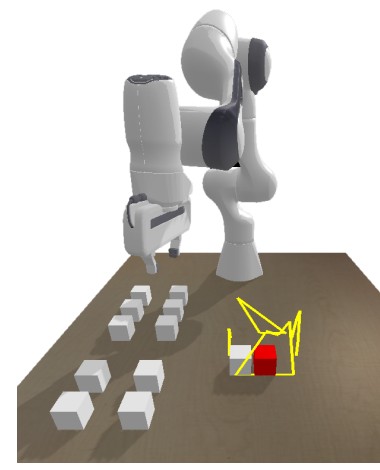

Figure 1: Simulated assembly environment with a 7-DoF manipulator and two sets of blocks: the unplaced white ones on the left and the "base" red block with some already placed blocks by the robot in the centre. The yellow silhouette denotes the target shape. The goal is to place the available blocks to fill the overall 3-dimensional target shape.

methods for Task and Motion Planning (TAMP) [9], and RL [10] to address the challenges of combinatorial optimization over high-level action sequencing and low-level motion generation.

In [11, 12] a TAMP method is proposed for robotic architectural construction and extrusion, requiring though exact domain specifications to find an assembly sequence; TAMP with logic-geometric programming is proposed by [13] where the focus lies in optimizing the structure's stability. In [14], the authors propose a hierarchical planner using hybrid dynamics models for the "toy-airplane" assembly task. The authors of [15] optimize the assembly sequence of complex interlocking blocks through RL assuming a single fixed design. A method based on neuro-symbolic planning is proposed by [16] for learning to predict sequences of actions for stacking. The authors of [17] propose an end-to-end approach for sequential pick-and-place tasks using shape correspondences and learning to assemble by collecting demonstrations from a human operator disassembling. Conversely, we *learn2assemble* arbitrary designs from scratch, learning both the sequence of actions and discovering goal positions per building block.

Autonomous assembly appeals to the machine learning community due to its combinatorial complexity, which, depending on the structure's size and availability of building blocks and their possible combinations [18], can by far surpass the state-action combinatorics of problems like chess and Go [19, 20]. The relational representation [21, 22] and generalization power of GNNs was thoroughly explored in solving combinatorial optimization tasks [23], with successful applications in 2D assembly [24] when combined with RL. In robot learning, earlier works use deep RL for short-horizon challenges like peg-in-a-hole [10, 25, 26]. In [27] the use of structured representations in model-free RL is proposed to induce inductive biases in different stages of a curriculum for executing assembly tasks of building towers of variable heights. While the general motivation of this work is close to ours, [27] does not learn the sequence of the building blocks for constructing arbitrary structures, but it is essentially a goal-based method for learning pick-and-placing manipulation with known goal object positions. In a similar direction, the authors of [28] use learning from demonstrations to train two GNNs, one that selects objects in the scene and another one that selects a suitable goal state from a set of possible goal positions.

## 2 Proposed Method

The main research question we pose in this work is *"how can a robot perform combinatorial assembly tasks of abstract architectural designs?"*. Fig. 1 depicts a typical setup containing a 7DoF manipulator robot, a varying number of building blocks, and one possible architectural target design. We formalize the task of autonomous construction of abstract target shapes given some available building blocks as a Markov decision process (MDP) [29]. As illustrated in Figs. 1 & 4 we can build one- up to four-sided structures. The overall desired structure is given by each side's target topology, which is in turn defined by the coordinates of the target points spanning the area to be filled (see illustrative example in Fig. 2). The desired shape is thus parameterized by the set of target points $\mathcal{S}_T = \{\mathbf{x}_{T_i} | i \in N_i\}$ (yellow).

The state also contains a set of blocks that are already placed in the scene $\mathcal{S}_P = \{\mathbf{x}_{P_j} | j \in N_j\}$ (red) and a set of unplaced blocks $\mathcal{S}_U = \{\mathbf{x}_{U_k} | k \in N_k\}$ (gray) that need to be added. The vector $\mathbf{x} \in \mathbb{R}^5$ includes the 3D position and two booleans indicating the element's properties (placed/unplaced, target point/building block). The state $s_t$ at any time step $t$ is thus given by the three sets $s = (\mathcal{S}_T, \mathcal{S}_P, \mathcal{S}_U)$ and has $N = N_i + N_j + N_k$ elements. The objective is to use the available blocks to construct a stable structure that fills the desired target design. We use a discrete action space and define five relative placement actions, i.e., any unplaced block can be put on top, on the left, on the right, behind, in front w.r.t. any already placed block. In the robot experiments, however, the action space is augmented to also allow the specification of the manipulator's orientation during grasping and placing.

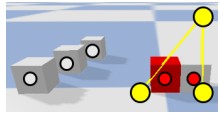

Figure 2: Setup with a one-sided triangular target shape defined by three points (yellow), three unplaced blocks (gray) and two placed blocks (red).

For most experiments, we choose over two possible grasping and placing orientations, resulting in four grasp-place combinations. Together with the placement actions, this yields $N_a = 4 \times 5 = 20$ actions when placing one block w.r.t. another. Since the possible actions $a_t$ depend on the number of placed and unplaced blocks, we end up with a time-varying action space of size $N_j \times N_k \times N_a$. We assign a positive reward $r(s_t, a_t)$ on all actions that increase the filling of the target structure while preserving its stability. Given the problem's combinatorial complexity of discovering the optimal high-level action plan that will allow the assembly of complex 3D structures which are stable and kinematically feasible, we decompose it into three main problems: (i) finding an expressive representation for our state space, (ii) decoding states into meaningful actions, and (iii) ensuring stability and kinematic feasibility. Our research provides a thorough study of how we can tackle these problems, and we provide a novel 3D assembly algorithm with the robot-in-the-loop through combining a learned multi-head attention (MHA) representation with MCTS.

### 2.1 Multi-head attention graph representation

Graph-based representations [30] are an effective tool when dealing with combinatorial problems [31, 32, 33]. Compared to classical satisfiability solvers, their main advantage lies in their real-time capabilities, while their architectural properties allow generalization to problems of different sizes in contrast to most standard neural network architectures, which operate on fixed-size inputs and outputs. As those are essential properties for learning to assemble structures of combinatorial complexity, we will introduce below our proposed GNN model that is inspired by the combination of graphs and attention [34, 35]. GNNs receive as input a graph $G = (\mathcal{N}, \mathcal{E})$, and return a high-level encoding over nodes and edges to be further exploited for deciding which action to take. In our case, the set of nodes $\mathcal{N} = \{\mathcal{S}_T, \mathcal{S}_P, \mathcal{S}_U\} = \{n_i\}_{i=1..N}$ is given by the current state, and the connectivity information $\mathcal{E}$ is defined as a matrix of size $N \times N$. If there is an edge connection between nodes $n_i$ and $n_j$ entry $\mathcal{E}(i, j)$ equates to 1, otherwise it is 0.

Attention mechanisms [34] were introduced in GNNs [35] to enable nodes to attend over their neighbours' features and learn different weights for different nodes without requiring costly matrix operations. We want to exploit this flexibility as solving the assembly task necessitates drawing connections on multiple levels, i.e., between nodes of the same type to encode the already existing structure or the target shape, as well as between all nodes to come up with a meaningful representation for action-decision. In the following, we will introduce the proposed MHA architecture, that naturally reflects the necessary multi-level decision process of the assembly task, and has proven to be effective when combined with policy search for solving combinatorial problems, like the travelling salesman [32]. In the first step of MHA, the initial node embeddings $n_i^{(0)} = \mathbf{x}_i$ are projected into a higher dimensional space by

$$n_i^{(1)} = g(n_i^{(0)}) = \text{ReLU}(\text{FC}(n_i^{(0)})), \tag{1}$$

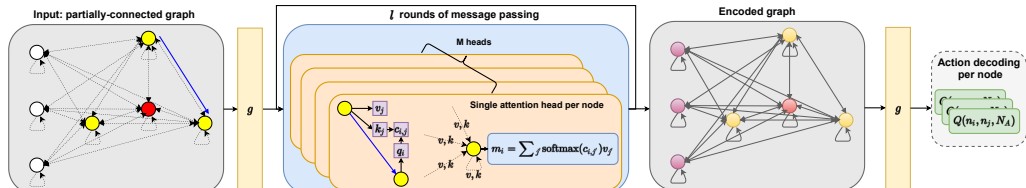

Figure 3: GNN architecture illustration, mapping from the input graph to Q-values. The coloring follows Fig. 2. After an initial projection into a higher dimensional space follow $l$ rounds of message passing using MHA. This results in an encoded version of the graph, which is then exploited for action selection.

using a fully-connected (FC) layer followed by a rectified linear unit (ReLU) activation function. Note that the function $g$ will be used repeatedly as we progress with the graph update, yet, assuming different weights on each further appearance. Next follow $L$ rounds of message passing, i.e.,applying (2) $L$ times to obtain the final embedding for each node $i$ according to

$$n_i^{(l)} = h(g(h(\text{MHA}(\mathcal{N}^{(l-1)}, i)))), \tag{2}$$

with a skip connection layer $h(f(x)) = x + f(x)$, the current round of message passing $l$, all node embeddings from the previous round $\mathcal{N}^{(l-1)}$, and a MHA mechanism introduced below. As illustrated in Fig. 3, for a single out of the $M$ attention heads with index $o$, we first compute three values – the key $k$, query $q$, and value $v$ – using three different weight matrices $W_{k,o}, W_{q,o}, W_{v,o}$, respectively ($k_{i,o} = W_{k,o} n_i^{(l-1)}$, $q_{i,o} = W_{q,o} n_i^{(l-1)}$, $v_{i,o} = W_{v,o} n_i^{(l-1)}$). Multiplying the key and query of all nodes results in a compatibility score $c_{i,j}$ for the $i^{\text{th}}$-to-$j^{\text{th}}$ node connection

$$c_{i,j,o} = \begin{cases} \frac{1}{d} q_{i,o}^T k_{j,o}, & \text{if } \mathcal{E}(i,j) = 1, \\ -\infty, & \text{otherwise,} \end{cases} \tag{3}$$

with $d$ a normalizing constant. From this score, we can then compute the attention weights using a softmax $a_{i,j,o} = \frac{e^{c_{i,j,o}}}{\sum_{j'} e^{c_{i,j',o}}}$ to aggregate the values for each node, resulting in the output message $m_{i,o} = \sum_{j'} a_{i,j,o} v_{j,o}$. A weighted sum of all messages yields the final result of the MHA module

$$\text{MHA}(\mathcal{N}^{(l-1)}, i) = \sum_{o=1}^{M} W_{m,o} m_{i,o}, \tag{4}$$

with weights $W_{m,o}$ controlling the influence of each one of the single attention heads. Fig. 3 depicts a simplified example of the encoding performed by the MHA architecture.

## 2.2 Learning assembly policies

MHA-GNNs can only unfold their potential when combined with algorithms that refine their weights to form expressive representations exploited herein for action selection (Fig. 3). The GNN should thus be shaped based on the reward signal, resulting in a RL setup with the goal of obtaining performant policies. This powerful combination results in agents that (i) can be applied to different problem instances due to the representation's flexibility, (ii) are reactive, despite the problem's combinatorial complexity, and (iii) can be trained directly in simulation environments which include the nonlinearities of the robot and the contacts. Due to the problem's combinatorial complexity and its discrete, time-varying action space, we use model-free Q-learning, [36] which has been successfully applied to complex tasks such as playing Atari games from images. Moreover, we investigate its integration with model-based planning, as the addition of search can counteract the overoptimism of the Q-function approximation and result in more robust behaviour [19, 37].

**Action Decoding.** The encoded graph representation from Sec. 2.1 needs further processing to decide on the next action to take. As actions are defined relatively between unplaced and placed blocks (i.e., nodes) we can directly assign a value to all available actions

$$Q(n_i, n_j, N_a) = g\left(n_i^{(l)}, n_j^{(l)}, \text{FC}\left(\frac{1}{N} \sum_{j'} n_{j'}^{(l)}\right)\right) \qquad \forall n_i \in \mathcal{S}_\mathcal{U}, n_j \in \mathcal{S}_\mathcal{P} \tag{5}$$

with the total number of $N$ nodes in the graph. Note that the Q-value does not only depend on the two nodes' embedding, but also on a global feature based on averaging over all embeddings. As all the operations are defined over the set of nodes, this encoding-decoding architecture can seamlessly generalize to different problem sizes, i.e.,different number of blocks or target shapes.

**DQN:** For our setting, we define the loss function as the smooth L1 loss between the current action-value estimate of the GNN, noted as $Q(n_i, n_j, a_s)$ and the value obtained from the rollouts using a

target network, noted as $Q_T$, i.e., $\hat{Q}(n_i, n_j, a_s) = r(s_t, a_t) + \gamma \max_{n_i \in \mathcal{S}_U, n_j \in \mathcal{S}_P} Q_T(n_i, n_j, a_s)$ with discount factor $\gamma$ and the selected action $a_s \in N_a$.

**MCTS:** MCTS [38] has proven effective for solving tasks with discrete action spaces [39], hence, suitable for our problem. Contrary to model-free RL, MCTS relies on a model of the environment to perform tree search for action selection and does not require any form of function approximation. However, in scenarios with significant branching factors or expensive model evaluations, the time complexity quickly increases, and the use of MCTS becomes intractable. Consequently, [19, 20] exploit Q-learning for estimating the value of the nodes in MCTS and show impressive results in solving tasks of great complexity, like the game of Go. Also, in the context of accelerating and generalizing skill-learning, the combination of learning and planning has received increasing attention [20, 37, 40, 41]. In this work, inspired by these advantages, we explore different algorithmic variations of the interplay between Q-learning and planning, as described below. Pseudocode for all variants is given in Appendix B.

**1) DQN+MCTS:** Inspired by [19], we propose a variant of AlphaGo, that uses a pretrained Q-network as a prior for evaluating the leaf nodes of MCTS, leading to increased efficiency. As we seek to effectively combine learning and planning for autonomous assembly with the robot-in-the-loop, we use small search budgets for which the prominent solution for MCTS expansion - the exploration strategy of UCT [19, 42] - is unsuitable because it becomes over-optimistic [43]. Therefore, we use an $\epsilon$-greedy expansion strategy during search, that allows better exploration than UCT.

**2) Q-MCTS:** We follow a generalization of Q-learning using samples based on planning. As in [20, 37], we combine DQN with MCTS during training to perform an informed exploration and collect good samples through search. However, as motivated previously, due to small search budgets, our approach uses an $\epsilon$-greedy expansion strategy over unexplored states, instead of UCT. Let $Q_S(s_t, a_t) = Q_S(s_t, a_t) + r(s_t, a_t) + \gamma \max_{a'_t} Q(s'_t, a'_t)$ be the updated value of an expanded node, where $s'_t$ is the new state arriving during search. The resulting Q-MCTS methods augments the DQN learning objective with a cross-entropy loss defined on the values explored during search, i.e., $\textbf{loss} = -\textbf{softmax}(Q_S(s_t, a_t))^T \log(\textbf{softmax}(Q(s_t, a_t)))$ which is intended to regularize and improve the Q-function estimate based on the experience collected while searching.

**3) $\epsilon$-MCTS:** This implementation follows [37] more closely. The main differences w.r.t. the Q-MCTS policy are that first, there is an $\epsilon$-greedy decision on whether to do search or uniformly sample a random action, whereas Q-MCTS always conducts search. Secondly, during search, $\epsilon$-MCTS follows the UCT expansion strategy. The Q-learning objective remains the same as with Q-MCTS, however, the cross-entropy loss is computed only on the samples where search has been conducted. In essence, the method's difference is in the way of collecting the model-based samples.

## 2.3 Integrated learning and planning for robotic assembly

All previous components can now be combined to obtain an algorithm capable of training agents to build desired target shapes, i.e., a robot capable of abstracting the sequence of actions for building arbitrary stable target structures from individual elements while executing feasible actions in its workspace. Note that the only control component we assume given is a point-to-point control strategy based on the robot's inverse kinematics. To enable this combined decision-making strategy that touches on the ground of TAMP literature, we propose the combination of the previously described graph representations and learning algorithms to provide a novel *learn2assemble* method. Briefly, a single step of a learning episode starts with selecting actions (with or without tree-search depending on the learning algorithm), i.e., the next object to be placed, the grasping pose, the goal position, and orientation. A path from the picking to the placing position is computed, and the robot executes the placement, for which it receives a reward. Consequently, we store the current graph's state, action, reward, and new state in the replay memory to be later used for learning.

## 3 Experimental Results

For evaluating the different components of the proposed method and their respective contribution, we designed specific experimental scenarios. We start with an investigation over graph architectures, continuing with the different learning methods in environments for 3D assembly without including a robot yet. Selecting the best settings from the previous tests, we experiment with the robot-in-the-loop for our final empirical evaluations of the proposed algorithm.

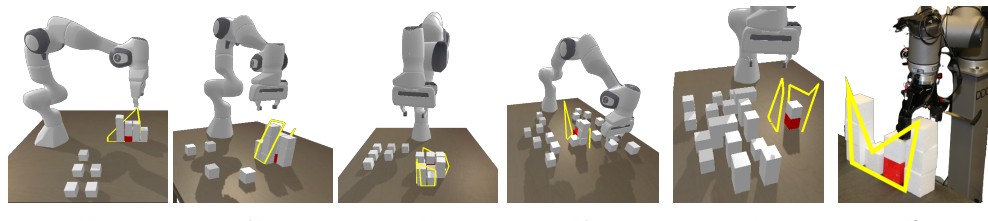

(a)       (b)       (c)       (d)       (e)       (f)

Figure 4: **(a)** Illustration of the single-sided 3D assembly environment, **(b)** the two-sided 3D assembly environment, **(c)** the four-sided 3D assembly environment with the robot-in-the-loop, **(d)** the two-sided 3D environment with the unplaced blocks placed at random, **(e)** the two-sided environment with different building blocks, and **(f)** the result of transferring a policy trained in simulation (a) to the real world and even a different manipulator.

Table 1: Comparison of different architectures on the single-sided environment without the robot and only one type of block. $R$ is the cumulative discounted return, $f$ the ratio of runs that ended with failure, i.e., the structure colliding, and $b$ the ratio of runs that ended without success and no more blocks remaining. The star(*) marks the environment where the agents were trained in.

| Method | 3-by-3 grid, 20-24 blocks* | | | 3-by-3 grid, 30-34 blocks | | | 4-by-4 grid, 20-24 blocks | | |
| --- | --- | --- | --- | --- | --- | --- | --- | --- | --- |
| | $R$ | $b$ | $f$ | $R$ | $b$ | $f$ | $R$ | $b$ | $f$ |
| MHA (FC) | **3.22** (0.04) | 0.01 | **0.15** | **3.44** (0.04) | 0.00 | **0.21** | **3.66** (0.05) | 0.18 | **0.41** |
| S2V (FC) | 2.36 (0.08) | 0.49 | 0.47 | 2.15 (0.10) | 0.08 | 0.87 | 2.75 (0.20) | 0.45 | 0.55 |

**Simulation environments.** The assembly environments are depicted in Fig. 4. We investigate, one-, two- and four-sided setups with each side of the target shape defined by the position of 3 (see Fig. 2), 4, or 5 target points. The number of target points and their locations are sampled randomly from grids of different sizes, ranging from 3-by-3 (i.e., the sampled target points can at maximum span an area of height and width of 3 times the cube's edge length) to 6-by-6. Due to the relative action space, every scene is always initialized with one initially placed element, marked in red, serving as the building base. For evaluating the assembly progression, we use depth cameras, placed on each side of the specified target structure. By projecting the target points into the images after each action, we obtain the change in the target shape's filling. The reward functions endorse actions that lead to improvement in the filling (cf. Appx. C.2). The construction process is finished, once the total coverage exceeds a threshold, whenever there are no more unplaced blocks available, or upon executing an invalid action, i.e., an action resulting in an unstable configuration, in destructing the current structure, or if it is kinematically infeasible. In all environments without the robot, the placing is done by directly specifying one of the five placement actions, resulting in a reduced action space of $N_a = 5$. If not stated differently we use partially connected graphs (see Fig. 3), inducing a stronger inductive bias on the structured representation compared to fully connected (FC) graphs (cf. Appx. A.3 & D.3). In the following tables, for evaluating the agent's performance, we report the cumulative discounted return $R$, the ratio of runs that ended with failure, i.e., upon an invalid action $f$, the ratio of runs that ended without success and no more blocks remaining $b$, as well as the mean number of actions conducted per run $\bar{a}$. The star(*) marks the agents' evaluation in the same setting as in training, while the rest are OOD experiments, i.e., exclusively evaluating the agents in settings with previously unseen target shapes or number of blocks. For more details, see Appx. C.

**Graph architectures.** We evaluate the proposed MHA representation against the commonly used Structure2Vector (S2V) architecture [24, 31] (cf. Appx. A.1) in a simple environment (see Fig. 4a) only considering one type of object but omitting the robot. The learning is conducted with DQN.

_Results._ As shown in the first column of Table 1, already in the original training environment, the MHA approach outperforms S2V significantly. The high rates of failure and exceeding the number of available blocks indicate that S2V cannot draw the connection between the target shape and the current structure. This might be due to S2V's different message passing, which cannot weigh the importance of different nodes as with the attention mechanism. When increasing the number of available blocks and the size of the structure to be built (columns 3 & 4), we see an evident advantage of MHA in handling OOD tasks. In Appx. D.2.1, we provide additional results when using only a single attention head, which confirm that using attention is advantageous, and MHA yields the best performance. We, thus, continue our experimentation using the MHA architecture.

**Learning algorithms.** To investigate the performance of the learning algorithms (Sec. 2.2), we will use the two-sided environment shown in Fig. 4b without the robot, thus using the reduced action space.

_Results._ Table 2 summarizes the results, starting without any search budget (i.e., no tree search) to evaluate the learned Q-functions. The DQN and $\epsilon$-MCTS agents perform similarly, with DQN slightly outperforming $\epsilon$-MCTS through lower failure rates across tasks. Moreover, DQN can solve

Table 2: Combining Q-learning and MCTS in the two-sided environment without the robot.

| Search Budget | Method | 3-by-3, grid 20-24 blocks* | | | 3-by-3 grid, 30-34 blocks | | | 4-by-4 grid, 30-34 blocks | | | 5-by-5 grid, 40-44 blocks | | |
|---|---|---|---|---|---|---|---|---|---|---|---|---|---|
| | | $R$ | $\bar{a}$ | $f$ | $R$ | $\bar{a}$ | $f$ | $R$ | $\bar{a}$ | $f$ | $R$ | $\bar{a}$ | $f$ |
| 0 | DQN | **3.21** (0.05) | 7.93 | 0.16 | **3.44** (0.08) | 8.65 | 0.17 | **3.61** (0.06) | 12.02 | 0.51 | **3.63** (0.08) | 13.66 | 0.93 |
| | $\epsilon$-MCTS | 3.18 (0.03) | 8.57 | 0.22 | 3.37 (0.10) | 9.30 | 0.30 | 3.54 (0.09) | 12.53 | 0.62 | 3.50 (0.13) | 12.75 | 0.95 |
| | Q-MCTS | 2.80 (0.15) | 7.33 | 0.51 | 2.89 (0.09) | 7.60 | 0.64 | 2.66 (0.08) | 7.47 | 0.92 | 2.34 (0.17) | 6.45 | 0.99 |
| 10 | DQN+MCTS | **3.47** (0.02) | 8.16 | **0.05** | **3.66** (0.03) | 8.96 | **0.06** | **3.96** (0.04) | 13.92 | **0.31** | **4.08** (0.09) | 17.33 | **0.87** |
| | $\epsilon$-MCTS | 3.21 (0.03) | 8.48 | 0.20 | 3.37 (0.11) | 9.10 | 0.31 | 3.60 (0.09) | 12.65 | 0.61 | 3.63 (0.16) | 13.16 | 0.93 |
| | Q-MCTS | 3.24 (0.04) | 8.56 | 0.27 | 3.39 (0.08) | 9.24 | 0.41 | 3.42 (0.07) | 10.99 | 0.76 | 3.41 (0.13) | 11.27 | 0.96 |
| 1000 | UCT | 0.54 | 4.00 | 1.00 | - | - | - | - | - | - | - | - | - |

the task with fewer actions. Contrarily, the Q-MCTS agents perform significantly worse. We believe that this difference in performance is due to the constant cross-entropy regularization in Q-MCTS from the beginning of the training, especially when search samples might be bad, while for the $\epsilon$-MCTS agent the regularization is only slowly added as training proceeds and less random actions are taken (search is better), which seems to be beneficial. Note that without the addition of search, increasing the number of blocks, as well as the target size, results in a quick increase of the failure rate for all methods. Adding a search budget of 10 already counteracts this trend, especially considering the DQN agent, where the rates of failure can be reduced for all the tasks by using DQN+MCTS at test time. For the Q-MCTS agent, the same trend is noticed, while the gains of the $\epsilon$-MCTS agent are marginal. This suggests that the $\epsilon$-MCTS agent is overoptimistic and seems to choose similar actions as to when not using search at all. Overall, combining MCTS with a pretrained DQN results in the best performance in our experiments. The last row of the table underlines the problem's combinatorial complexity, illustrating that performing pure UCT without any prior and a search budget of 1000 performs significantly worse for the simplest experimental setting (more details in Appx. D.4).

**Learn2assemble with the robot-in-the-loop.** Next, we evaluate the DQN agent combined with tree-search in our target environments, including the robot manipulator. The task's difficulty increases, as also the grasping and placing poses have to be specified while ensuring action feasibility by the robot and the structure's stability. We start with constructing single-sided designs (Fig. 4a) to illustrate the necessity of training with the robot-in-the-loop before evaluating the proposed method with multi-sided designs (Fig. 4b & 4c).

_Results._ We first compare two policies, one trained with, and the other without the robot using plain DQN without any search in a single-sided environment (Fig. 4a). The agent trained with the robot-in-the-loop outperforms the other one resulting in a significantly reduced failure rate of $15\%$ and consistently higher rewards (cf. Appx. D.5). This shows the necessity of including geometric planning during training for obtaining high-level decisions that are compatible with the low-level execution. It is thus not sufficient to only figure out where to place the parts; the kinematic constraints and robot motion have to be considered. In the subsequent experiments, we build 3D structures with the robot as shown in Figs. 4b & 4c. Our results in Tables 3 and 4 show that MCTS (search budget of 10) consistently improves performance in terms of higher returns and lower failure rate. In particular, the experiments demonstrate that our proposed pipeline can execute multiple sequential pick-and-placing actions, with a

Table 3: Comparing policies with and without tree search on the four-sided robotic environment.

| Method | 3 by 3 grid 10-18 blocks* | | | 3 by 3 grid 16-24 blocks | | |
|---|---|---|---|---|---|---|
| | $R$ | $\bar{a}$ | $f$ | $R$ | $\bar{a}$ | $f$ |
| DQN | 2.67 (0.06) | 6.91 | 0.30 | 2.55 (0.06) | 7.25 | 0.35 |
| DQN+MCTS | **3.08** (0.06) | 7.59 | **0.16** | **2.90** (0.07) | 8.00 | **0.20** |

maximum of 17 correctly placed blocks for the 5-by-5 grid in the two-sided environment and up to 22 correct placements in the four-sided environment, using DQN+MCTS with the robot-in-the-loop. Compared to the previous experiments, without the robot-in-the-loop, the failure rate is higher, indicating the task's increased difficulty, and the need of adding a soft-placing controller.

**Generalization w.r.t. randomized scenes.** To evaluate our algorithm's robustness w.r.t. changes in the scene, we transfer the previously trained policies and evaluate them in scenarios where the unplaced blocks are placed randomly around the structure to be built, as shown in Fig. 4d.

_Results._ Rows 3 & 4 of Table 4 reveal that the policies indeed generalize to these novel scenarios, as the percentage of unsuccessful experiments only increases at maximum by $11\%$ for the most complex scenario, compared to their performance in the original environment (rows 1 & 2). This confirms that our proposed method does not overfit the exact layout or geometry, but rather builds meaningful features that allow to successfully transfer the behaviour to scenes that are substantially different from those encountered during training.

Table 4: Comparing policies on the two-sided robotic environment. Rows 1&2 correspond to evaluating in the original environments (Fig. 4b), while rows 3&4 are the evaluation in randomly initialized scenes (Fig. 4d).

| Method | Environment initialization | 3 by 3 grid 10-14 blocks* | | | 3 by 3 grid 14-18 blocks | | | 5 by 5 grid 14-18 blocks | | |
|---|---|---|---|---|---|---|---|---|---|---|
| | | $R$ | $\bar{a}$ | $f$ | $R$ | $\bar{a}$ | $f$ | $R$ | $\bar{a}$ | $f$ |
| DQN | fixed | 2.16 (0.06) | 4.89 | 0.20 | 2.10 (0.05) | 5.27 | 0.24 | 2.54 (0.15) | 7.53 | 0.56 |
| DQN+MCTS | fixed | **2.41** (0.05) | 5.32 | **0.09** | **2.32** (0.03) | 5.64 | **0.15** | **3.19** (0.11) | 9.38 | **0.36** |
| DQN | random | 2.06 (0.10) | 4.93 | 0.24 | 1.88 (0.13) | 5.02 | 0.32 | 2.28 (0.11) | 7.52 | 0.64 |
| DQN+MCTS | random | 2.33 (0.06) | 5.38 | **0.10** | 2.14 (0.12) | 5.54 | **0.20** | 2.85 (0.14) | 9.31 | **0.47** |

Table 5: Evaluating the trained policies in the environments with multiple different objects available (Fig. 4e). The results in the first row correspond to using a modified environment without including the robot.

| Method | Environment w/wo robot | 5 by 5 grid 20-24 blocks* | | | 5 by 5 grid 30-34 blocks | | | 6 by 6 grid 30-34 blocks | | |
|---|---|---|---|---|---|---|---|---|---|---|
| | | $R$ | $\bar{a}$ | $f$ | $R$ | $\bar{a}$ | $f$ | $R$ | $\bar{a}$ | $f$ |
| DQN+MCTS | wo robot | **2.20** (0.02) | 5.05 | **0.06** | 1.89 (0.03) | 5.96 | **0.15** | 1.48 (0.09) | 7.27 | **0.27** |
| DQN+MCTS | w robot | 1.42 (0.05) | 3.53 | **0.21** | 0.77 (0.09) | 3.27 | **0.52** | 0.61 (0.08) | 4.38 | **0.61** |

**Generalization w.r.t. different building blocks.** We finally investigate our method's performance when using more complex objects (Fig. 4e). This makes the task significantly more difficult, as the agent has to not only learn each part's admissible grasps but also differentiate the building blocks to select and place the correct object type. All novel objects are a combination of primitive boxes which allows keeping the relative action space, i.e., placing an unplaced primitive box belonging to a larger object w.r.t. a placed primitive box results in moving the entire object. For the experiments without the robot, we adjust the action space to enable changing the object's orientation (cf. Appx. D.9). *Results.* The results in Table 5 illustrate the representation's flexibility and ability to successfully deal with the different building blocks. Despite the increased complexity, we achieve similar performance in the scenes without the robot as in previous experiments (Table 2). When training with the robot, we can still handle the task's complexity with a $\sim 80\%$ success rate in the simplest setting, but observe a drop in performance with a larger number of objects and bigger target shapes. While one cause for the performance drop is the setting's increased complexity, there are also several placement actions that would require 3D gripper orientation control allowing for a smooth insertion of the complex blocks. As our current top-down placing controller only offers planar orientation control, some placement actions cannot be executed appropriately resulting in increased failures.

**Remarks.** We have conducted extensive experiments to show the superiority of our proposed MHA-GNN approach for solving combinatorial assembly tasks with the robot-in-the-loop. The results demonstrate how strong inductive biases combined with attention can shape meaningful relational representations. When combined with deep Q-learning, this representation allows us to take decisions over long horizons despite an increasing action space. Adding search at test time improves performance across all experiments, demonstrating that the proposed method can generalize w.r.t. different target shape sizes, number of building blocks, and different scenes. The policies can also be transferred to the real world and to different manipulators, by initializing the simulation to mirror the real scene and executing the obtained actions both in simulation and reality (see Fig. 4f). While we show very promising results in combining learning high-level action decisions with planning geometric executions, we can see a combinatorial barrier in the decision-making that might be tackled with a more informed search in the graph space. Our experiments on using different objects underline the representation's flexibility, but also reveal current limitations in the definition of the action space, especially considering the robotic execution, which may be mitigated by enriching the action space through 6D grasping and placing. Moreover, several assemblies failed due to "rough" robot actions, meaning that a more sophisticated motion generator might be needed for finer placement.

# 4 Conclusion

We presented a new *learn2assemble* algorithm for learning autonomous robotic 3D assembly from scratch without prior knowledge of any task plan. For addressing the problem's combinatorial complexity while maintaining adaptability to different scenarios, we propose a graph-based multi-head attention representation that captures the spatial relationships between target construction designs and unplaced blocks, and is trained through deep Q-learning. The powerful representation forms the basis for our hierarchical controller that jointly conducts high-level learning over action sequences and goal specifications together with low-level path planning, ensuring the execution of long-horizon tasks. Our extensive experiments confirm the representation's effectiveness and show extrapolation to environments with previously unseen target shapes, larger numbers of available elements, and different object types. When combining the learned Q-network with MCTS with computationally tractable small search budgets, we manage to improve performance and reliability across all tasks. Notably, we resolve the sequential long-horizon character of the assembly task by including the robot-in-the-loop to decide over feasible grasps and placing actions that ensure the stability of the construction. Our algorithm manages to correctly build structures using up to 22 building blocks with good success rates. In the future, we want to extend the algorithm to allow for a richer set of 6D grasping and placing poses, learn fine-placing or even in-hand manipulation controllers on the low level, and investigate the implementation of an assembly/disassembly strategy, so that the robot can potentially re-use wrongly placed blocks or reconfigure existing structures.

**Acknowledgments**

The authors acknowledge the support from the Artificial Intelligence in Construction (AICO) grant by the Nexplore/Hochtief Collaboration Lab at TU Darmstadt. This project has also received funding from the European Union's Horizon 2020 research and innovation programme under grant agreement No. 640554, and the Emmy Noether DFG Programme iROSA with Grant No. 448644653. The authors would like to thank the reviewers and metareviewer for their useful comments that helped to substantially improve the paper, and Prof. Dr. Oliver Tessmann from the Digital Design Unit (DDU) at the Department of Architecture at TU Darmstadt, for supporting us with the robotic experiments.

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
