# OpenReview forum: "Learn2Assemble with Structured Representations and Search for Robotic Architectural Construction"
_robot-learning.org/CoRL/2021/Conference — CoRL2021 Poster_

### Official Review · Reviewer_2DWg · 2021-07-13

**Originality:** Good
**Technical Quality:** Good
**Clarity Of Presentation:** Good
**Impact:** 3

**Recommendation:**

Weak Accept: I recommend accepting the paper, but will not argue for my recommendation if the majority of other reviewers have a different opinion.

**Summary:**

The paper proposes a method to learn structure assembly policies via RL where using Graph Neural Networks. The input to the network is a graph representation of the scene, where each node represents an object (rectangular building block). The output of the network is the Q-value for different actions. The task is specified by stacking blocks to fill up a target volume. The authors evaluate a variety of RL algorithms in simulation and achieve a success rate of 85% in the best case and 49% in the hardest case (harder cases have more blocks and larger structures).

**Issues:**

Please see the Strengths and Weaknesses section.

**Reviewer Expertise:**

Good: General knowledge of the area

**Strengths And Weaknesses:**

Strengths:
-	The proposed method of incorporating graph-based inductive bias in the construction task is suitable.
-	The experiments and results are thorough.
-	The analyses provide insights on comparing various graph-based representations, like multiheaded-attention and structure-to-vector.
Weaknesses
-	My main criticism of the paper is that it is poorly written, or at least, difficult to parse. The wording is dense both from a language and visual perspective. There is some key information I was unable to easily find in the paper. I list them as the following questions.
-	Is the task (target build volume) encoded in the observations, or do we need to train a new Q function for a new target?
-	One of the main benefits of GNNs is its ability to generalize across different numbers of nodes and edges. In the case of this paper, can the GNNs trained on small number of objects generalize to bigger ones?
-	Is it the case that the current action encoding limits newly placed blocks can only be placed next to previously placed blocks? If so, it is unclear how the action encoding can be made more flexible to avoid this limitation when needing to build structures that have hollow volumes.
-	The task volume visualization is also difficult to understand, both in the paper and in the videos. I think it may be clearer to represent target structures with semi-transparent volumes rather than outlines. It is also unclear how a set of rectangular blocks is supposed to construct some of the visualized structures that contain slanted sides.
-	The improvement in performance when training with robot-in-the-loop may come at a cost of overfitting to the robot’s geometry and kinematics. How well would this transfer to new robots or even the same robot with a different end-effector?
-	It would be helpful to see an additional baseline comparison against using MCTS with a simple hardcoded heuristic (value) function. This will illustrate the additional information learned by DQN.
-	While many experiments were performed, it is difficult to understand their significance and what solid conclusions should be drawn from them. Clearer discussions on this would have been helpful.


**Summary Of Recommendation:**

My recommendation of weak reject is primarily due to the lack of clarity in the paper writing, which made it difficult for me to clearly assess the paper’s technical merits and contributions.

**Update** I have updated my recommendation to weak accept after the authors' revisions.

---

> ### Author Response · Authors · 2021-08-27
> **Response to reviewer 2DWg (1/2)**
>
> We thank the reviewer for the detailed review. We have worked on rephrasing parts of the paper to make it clearer and easy to follow. Regarding the specific questions of the reviewer, we provide below answers, while we are also running additional experiments as requested, which we will update on a follow-up reply.
>
> >Is the task (target build volume) encoded in the observations, or do we need to train a new Q function for a new target?
>
> The target shape is encoded as part of the graph. We have updated the introduction of Sec. 2 and added Fig. 2 to underline the description of the environment.
>
> >As illustrated in Figs 1 & 4 we can build one- up to four-sided structures. The overall desired structure is given by each side's target topology, which is in turn defined by the coordinates of the target points spanning the area to be filled (see illustrative example in Fig. 2).
>
> This is also depicted in the visualization when introducing the multi-head attention architecture (see Fig. 3), where the yellow nodes represent the vertices of the goal shape. The multi-head attention graph neural network encodes the spatial information between target shape and available building blocks, and computes the Q values for all available actions between every unplaced and placed block.
>
> We further clarify our method’s generalization capabilities in Section 2.2, “action decoding” at the bottom of page 4:
> >As all the operations are defined over the set of nodes, this encoding-decoding architecture can seamlessly generalize to different problem sizes, i.e.,different number of blocks or target shapes.
>
>
> >One of the main benefits of GNNs is its ability to generalize across different numbers of nodes and edges. In the case of this paper, can the GNNs trained on small number of objects generalize to bigger ones?
>
> The GNNs can generalize to both a greater number of building blocks and to different sizes of target shapes compared to those used during training, but also to different types of building blocks that consist of primitive blocks. These results are provided in Tables 1-6 where all results, not denoted by *, are testing settings that are unseen during training. Hence, we can argue that the GNN effectively generalizes as the reviewer states accurately.
>
>
> >Is it the case that the current action encoding limits newly placed blocks can only be placed next to previously placed blocks? If so, it is unclear how the action encoding can be made more flexible to avoid this limitation when needing to build structures that have hollow volumes.
>
> Depending on the complexity of the objects, we have to either augment the action space, or define actions that are module specific. We are currently running new experiments with more complex objects (tetris-like blocks). We will soon update the paper with the new results. However, to include more abstract shapes, we need a different encoding. In future work, we will investigate  the possibility of encoding each building block as a graph which in turn encodes its pose and all potential grasping poses. Such a solution would require a higher -level of abstraction on top of the proposed multi-head-attention graph representation, that we will be thrilled to explore afterwards. This work, on the other hand, tries to address the combinatorial complexity of 3D robotic assembly, and we propose a holistic approach, which we plan to extend in the future.
>
> For building hollow volumes, we would have to modify our way of representing the target shape. In particular, we would have to define additional attributes to the node’s embedding in order to identify the parts that should not be filled. Moreover, when wanting to build hollow shapes we would have to provide two initially placed elements (red blocks) in different locations. Starting from those two positions we could potentially construct hollow shapes while keeping the current action space.

---

> > ### Author Response · Authors · 2021-08-27
> > **Response to reviewer 2DWg (2/2)**
> >
> > >The task volume visualization is also difficult to understand, both in the paper and in the videos. I think it may be clearer to represent target structures with semi-transparent volumes rather than outlines. It is also unclear how a set of rectangular blocks is supposed to construct some of the visualized structures that contain slanted sides.
> >
> > The target shapes are currently arbitrary structures that we randomly generate during training. Indeed, we currently cannot perfectly cover structures with slanted sides with the current building blocks. In the future, we want to encode the mesh of various building blocks, and extend the algorithm to a higher level of abstraction (reason on object shape and then reasoning on the sequencing of the blocks).
> >
> > >The improvement in performance when training with robot-in-the-loop may come at a cost of overfitting to the robot’s geometry and kinematics. How well would this transfer to new robots or even the same robot with a different end-effector?
> >
> > Regarding the robotic geometry, there is no way to overfit the geometry of the robot, as the robot only operates in the task space. As long as the building blocks and the assembly area is reachable by a robotic manipulator, the geometric motion execution will not be affected.
> >
> > >It would be helpful to see an additional baseline comparison against using MCTS with a simple hardcoded heuristic (value) function. This will illustrate the additional information learned by DQN.
> >
> > Thank you for pointing this out. We are unsure about practical ways to obtain a hardcoded heuristic value function on the considered tasks. Nevertheless, we believe that the most representative example of the impact of DQN in approximating the Action-value function during search arises from the plain use of UCT search, where the Q-function is not learned, but approximated during the expansion and back-up of the tree, according to the Bellman back-up. In the last line of Table 3, we report the results of solely using UCT, for the simplest case of constructing a shape using up to 24 blocks. We can see that even with a much bigger tree (search budget of 1000), UCT cannot match the performance of the DQN agent, and of those combined with search. Moreover, having larger search budgets makes the problem computationally inefficient. This clearly illustrates another benefit of combining DQN+search as the method is already effective with small budgets which is crucial for robotics applications.
> >
> > >While many experiments were performed, it is difficult to understand their significance and what solid conclusions should be drawn from them. Clearer discussions on this would have been helpful.
> >
> > We apologize if the experimental section is not clear. We are working on updating this part with additional experiments, and a more detailed discussion on the results. We will update you accordingly.

---

> > > ### Comment · Reviewer_2DWg · 2021-08-31
> > > **Rebuttal Response**
> > >
> > > I would like to thank the authors for their detailed response and revisions. I have updated my recommendation to weak accept.

---

> > > > ### Author Response · Authors · 2021-09-01
> > > > **Thank you**
> > > >
> > > > Thank you for increasing your score.

---

> ### Author Response · Authors · 2021-08-31
> **Information on final version of the paper and added experiments**
>
> We have updated the entire paper substantially and hope that the new version is clearer and easier to follow.
>
> In particular, we have substantially updated the experimental section by moving some experiments to the Appendix, adding new results as well as discussing our findings more appropriately at the remarks paragraph at the end of section 3 of the paper.
>
> In particular, we have updated our experimental results (as suggested by other reviewers) and added a comparison of the graph architectures to the single-head attention mechanism as introduced in Appx. D.2.1 and illustrated in the videos on our website (https://sites.google.com/view/learn2assemble). The results show that the proposed multi-head attention graph architecture is indeed very effective as it outperforms the commonly used Structure2Vector architecture as well as the single head attention approach.  As single head attention also performs better than Structure2Vector, we reason that using the attention mechanism for the message passing is beneficial as it allows learning different weights. However, multiple attention heads are necessary to encode the increased relational complexity in the assembly scenes.
>
> Secondly, we have provided an evaluation of a trained agent with the robot-in-the-loop in a scene with randomized initial block positions (as compared to the structured setting of training), where we show the flexibility of the multi-head-attention graph representation to encode spatial relations without overfitting to a specific setting (see Table 4 as well as the accompanying video). This also shows that the agents can indeed generalize to novel scenes where the parts are initialized differently. Our proposed method is thus not bound to be only used in exactly the same scenes as it has been trained in.
>
> Thirdly, we have provided a more complex experimental scenario where we use different objects to assemble target shapes (see Fig. 4e and Table 5). We demonstrated that the proposed MHA-GNN-RL algorithm manages to encode the spatial relations between various complex shapes and the desired target shape, as the performance of the agent trained without the robot, is the same as with the simple object-set. When training with the robot, we get a performance of up to ~ 80% success rate (94% without the robot), when evaluating on the same number of provided building blocks as with training. However,  when we add more objects the performance reduces to ~ 40% (~ 60% without the robot) with ~ 30 available blocks. Please note that in all these experiments the generation of initial objects in the scene is random, and we do not know whether the structure could actually be built effectively with the available blocks.  We attribute this performance to the limited action space of the robot (we only control planar orientations), and the fact that certain placements would need a fine insertion policy rather than a “rough” placing that leads to unstable structures. We also think that adding 6D grasps would improve the performance in these more complicated settings. We conclude from these experiments that the learned representations can indeed also generalize in scenarios with different available building blocks but do note that the current definition of the action space limits the method’s performance in the scenarios including the robot. Overcoming these limitations will be subject to future work.

---

### Official Review · Reviewer_RVUr · 2021-07-22

**Originality:** Very Good
**Technical Quality:** Very Good
**Clarity Of Presentation:** Excellent
**Impact:** 4

**Recommendation:**

Strong Accept: I recommend accepting the paper and will argue for my recommendation even if other reviewers hold a different opinion.

**Summary:**

The paper studies the problem of 3D assembly, more specifically having a 7 DOF robot arm stack blocks into a target shape. The paper aims to maximize combinatorial generalization for this task, by studying different choices of graph representations (through the use of GNNs and multi head attention) and monte carlo tree search algorithms (combined with Q-learning). The experiments show that a partially connected architecture for graph representation learning is superior than the fully connected version, and this is then tested in conjunction with DQN+MCTS algorithm for the best results.

**Issues:**

See above.

**Reviewer Expertise:**

Poor: Limited knowledge of the area

**Strengths And Weaknesses:**

Strengths:
- The flow of the paper is great, starting with a clear goal in mind, proposing very reasonable choices for each component of the pipeline with just the right amount of explanation, and testing the different components (representation and algorithm) separately.
- The experimental section is thorough and clear. I really appreciate the targeted experimentation on each of the components, as it makes the results more reproducible and scientifically valuable for future work. The design of the novel benchmarking environment is also a contribution.

Weaknesses:
- It may be helpful to include some visualizations of the DQN+MCTS search procedure for a small toy example (e.g. with possible future states/actions and their estimated Q-functions). The toy example can also help better distinguish the different search methods described in section 2.2.

**Summary Of Recommendation:**

The problem of 3D assembly is obviously important, and the paper provides valuable contributions in terms of empirical evaluation of graph representation learning and Q-learning / MTCS applied to this task. The presentation is clear and the experiments are thorough and provides potentially valuable insights for future work.

---

> ### Author Response · Authors · 2021-08-27
> **Response to reviewer RVUr**
>
> We thank the reviewer for the positive feedback. We are very happy that you acknowledge the scope of the paper and our contributions.
>
> As requested, we have provided an illustration of the way the method conducts search in the supplementary material in Appendix D. 7, illustrating the different actions that have been explored during search, which include selecting different blocks and different grasps.
>
> Moreover, we will provide the codebase for benchmarking publicly upon paper acceptance, for other researchers to experiment and build their own assembly tasks.
>
> Please note that we have updated the paper, and we are currently running  more complex experiments as requested by other reviewers.

---

> ### Author Response · Authors · 2021-08-31
> **Information on final version of the paper and added experiments**
>
> We have integrated additional experiments regarding single-attention vs. multi-head attention in Appx D.2.1., showcasing the increased efficiency of the multiple attention heads to encode the complex spatial relations between placed objects, unplaced ones and target shapes. Moreover, in Table 4 of the paper (and the accompanying video) we added an evaluation of the agent with the robot-in-the-loop in scenes that are initialized randomly, as opposed to the “structured” object setting, where we can see that the generalisation of the spatial representation that does not overfit a specific geometry in the scene.
>
> Finally, we performed experiments with a more challenging object-set, for which we are reporting results both with and without the robot in Table 5 (you can also find videos on our website https://sites.google.com/view/learn2assemble). When training without the robot, we also incorporate the decisions over rotations. We can see that despite the increased action space and an increasing number of blocks, the performance remains comparable to the simple setting when only using simple boxes. When training with the robot, we have experienced a performance drop, which is attributed to the limited action space of the robot (we only control planar orientations), and the fact that certain placements would need a fine insertion policy rather than a “rough” placing that leads to unstable structures. Still, our results are promising and provide an interesting direction for future research for improving the low-level robot controller to allow us to do successful high-level reasoning with our proposed MHA-GNN-RL algorithm. We have also added an appropriate discussion of our findings at the remarks paragraph at the end of section 3 of the paper.

---

> > ### Comment · Reviewer_RVUr · 2021-09-02
> > **Thanks**
> >
> > Thank you for the response, they address the concerns I had. I still find the experiments interesting, although I'm not familiar with the literature.
> >
> > I also read the other reviews, and I think the simplicity of the environment is a valid concern of the other reviewers, since the block stacking is done in simulation. That said, I also don't have the background to judge if real-world block assembly is too much to ask for. In light of this, I've lowered my reviewer expertise score.

---

### Official Review · Reviewer_a8Bf · 2021-07-23

**Originality:** Fair
**Technical Quality:** Good
**Clarity Of Presentation:** Good
**Impact:** 3

**Recommendation:**

Weak Reject: I recommend rejecting the paper, but will not argue for my recommendation if the majority of other reviewers have a different opinion.

**Summary:**

This paper proposes a data-driven method for robotic assembly by leveraging structured state representations in conjunction with learning-based algorithms to effectively solve tasks. Specifically, the state representation is based on an graph neural network that incorporates attention mechanisms, and the learning algorithm is a hybrid model-based and model-free reinforcement learning algorithm. Experiments on a simulated assembly environment demonstrate that the proposed method can construct a variety of structures, outperforming variants with alternative state representations and learning algorithms.

**Issues:**

See the weaknesses list. In addition:
* Many of the experiments are conducted without the robot — what is the motivation behind this? Perhaps the reason is that introducing the robot may introduce confounding factors in the analysis, in which case the paper should clearly justify why this may be the case
* Numerous citations to GNN-based works are missing, including but not limited to (1) Relational deep reinforcement learning, by Zambaldi et al, (2) Relational inductive bias for physical construction in humans and machines, by Hamrick et al, (3) Contrastive learning of structured world models, by Kipf et al

**Reviewer Expertise:**

Good: General knowledge of the area

**Strengths And Weaknesses:**

Strengths:
* The method is explained in a thorough yet concise manner
* The experiments perform a systematic study of the state representation and learning algorithm
* The experimental analysis is generally thorough: when there is a positive or negative result, the paper accompanies that result with a discussion of the underlying reasons behind the result

Weaknesses:
* The environment seems a bit simple and artificial —  the blocks are of the same shape and size and the unplaced blocks are laid out in a regularly spaced manner. Intuitively, it seems that the robot can choose to pick up any of the unplaced blocks, as they all have the same functionality. This makes the problem seem artificially simple. Furthermore, the target configurations for the blocks seem quite arbitrary, how do they translate to real-world structures that are of interest in assembly/construction applications?
* The learning algorithms adopt elements from MCTS, a model-based algorithm. This raises the question, where does the underlying model come from? If making model predictions is based on having access to a simulator rather than a learned model, then this approach is unlikely to scale up due to the expensive computational load of simulators
* Experiments can still go deeper, ideally answering the following questions: (1) how much of a difference does multi-head attention make compared to single-head attention? (2) what forms of generalization can the proposed method achieves and what forms can it not achieve? (3) are the representations learned by the proposed framework useful for transfer to semantically related tasks?

**Summary Of Recommendation:**

This paper is well-written, clear, and the experimental analysis is generally thorough. Ultimately, my primary concern is with this paper’s overall motivation and message. The paper positions itself as an applications-focused work, seeking to advance robotic automation for assembly and construction. Yet the experimental domain does not appear to be a convincing or inspiring medium through which to advance the state-of-the-art on these applications. On the other hand, the paper does not seek to make broader points about how its findings are useful to the general robot learning community. As a result, the impact of this paper (in its current form) appears limited. I recommend either revising the message of the paper to be more universal, or revising the experiments to demonstrate more impressive results in assembly / construction.

**Update after rebuttal** my recommendation stays the same. Please refer to my followup post for a detailed explanation.

---

> ### Author Response · Authors · 2021-08-27
> **Response to reviewer a8Bf (1/3)**
>
> We thank the reviewer for the thorough review and effort. In the following, we hope to respond to your concerns. Please note that we have updated the paper and the appendix (in the supplementary material), and we are running additional experiments on more complex settings, which we will add soon.
>
> >The environment seems a bit simple and artificial — the blocks are of the same shape and size and the unplaced blocks are laid out in a regularly spaced manner. Intuitively, it seems that the robot can choose to pick up any of the unplaced blocks, as they all have the same functionality. This makes the problem seem artificially simple. Furthermore, the target configurations for the blocks seem quite arbitrary, how do they translate to real-world structures that are of interest in assembly/construction applications?
>
> Regarding the reviewer’s concern on the simplicity of the setup, we want to point out that the task of robotic assembly, where the objective is to build abstract target designs, is one of high combinatorial complexity, and it has not yet been explored in 3D robotic assembly context. Even with the simple setting, there is a bound on the performance of the different algorithms when increasing the number of available objects and the size of the area, or volume that needs to be covered (e.g., the combinatorics for placing one single block w.r.t. already 10 placed blocks in the structure has  $5\cdot2\cdot2\cdot10=200$ possibilities in our current setting). However, we are currently running and evaluating new experiments with an extended block set with tetris-like blocks, and we will update you soon.
>
> Regarding the robotic task, we had to ensure the reachability of the blocks by the robot. We would like to point the reviewer to the video in the supplementary material that showcases the execution. Looking carefully, one can see that the robot actually chooses the blocks in a smart manner (even if they are all the same). To cover the area further away, it chooses to use blocks closer to the area of interest. This makes sense as the distances of the building blocks to the current structure are encoded in the graph through the edges and those distances are taken into account by the multi-head attention mechanism. Also, first placing the blocks that are further away mitigates the risk of colliding with the already existing structure.
>
> To showcase the generalization power of the proposed representation, we are conducting experiments in which the primitive blocks are placed irregularly around the building area (but in the reachability of the robot). We will add these results soon, and notify you accordingly.
>
> For the target configurations, we confirm that they are arbitrary, as for now we just generated shapes that are defined by randomly sampling 3, 4, or 5 points. While this is not a plan of a real-world structure, the underlying idea of discovering the sequence of placement actions of building blocks in construction is actually a realistic problem, and we believe that our algorithmic solution attempts one step towards this direction. We also want to point out that the randomly sampled target shapes showcase the flexibility of our algorithm of being capable to adapt to a very wide range of target designs without requiring any retraining. In the future, we plan to transfer the logic of this algorithm in a realistic setting with an industrial robot. However, this requires more technologies to be integrated (e.g., accurate 6D pose estimation of multiple building blocks, etc.). Our main contribution is the design of a multi-head attention representation that can scale to solving combinatorial 3D assembly tasks with low-level action execution by a 6DoF robotic manipulator.

---

> > ### Author Response · Authors · 2021-08-27
> > **Response to reviewer a8Bf (2/3)**
> >
> > >The learning algorithms adopt elements from MCTS, a model-based algorithm. This raises the question, where does the underlying model come from? If making model predictions is based on having access to a simulator rather than a learned model, then this approach is unlikely to scale up due to the expensive computational load of simulators
> >
> > Our method conducts search using MCTS planning. We do not learn the model of the world, though this could possibly be encoded through the graph and allow us to also learn the dynamics of the robot and the building blocks. This is actually a future research direction. However, approximating the world model leads to compounding errors that affect the planning process, making it challenging to collect informative samples.
> >
> > On the other hand, offline planning with a simulator is a commonly used practice in robotics. As the proposed algorithm performs high-level action planning, even if the computational load is existent, yet it is manageable, and this holds for plenty of applications of robot-learning methods that require a combination of learning and planning. Importantly, the small search budget of the additional search when combined with the learned Q-function using the MHA-GNN architecture (acting as a prior) is very efficient computationally. Especially when compared to pure tree-based methods that require thousands of samples but still perform considerably worse (see for example the performance of using UCT with a search budget of 1000 without any prior in Table 2).
> >
> > >Experiments can still go deeper, ideally answering the following questions: (1) how much of a difference does multi-head attention make compared to single-head attention? (2) what forms of generalization can the proposed method achieves and what forms can it not achieve? (3) are the representations learned by the proposed framework useful for transfer to semantically related tasks?
> >
> > We thank the reviewer for the insights regarding our experiments. Regarding single vs. multi-head attention, we have added an experimental comparison in the supplementary material in Appendix D.2.1 and in the videos on our website (https://sites.google.com/view/learn2assemble).. The result shows that using attention results in more powerful representations as both, single-head and multi-head attention outperform the structure2vector representation. However, the results also reveal that multi-head attention (MHA) indeed outperforms the single-head approach. We attribute this to the fact that the multiple attention heads enable to come up with richer representations that better represent the complex interactions between all placed/unplaced blocks as well as the target elements.
> >
> > Regarding the generalization power of the proposed MHA-GNN, we evaluate and discuss this property in the experiments of Tables 1 and 2, where we tested the algorithm when handling more building blocks and larger target shapes compared to those seen during training, but also evaluate generalization to different types of objects (Table 5). The current version of our method can effectively generalize both to more building blocks and different shapes. In Tables 3, 4 and 5 we have also shown generalization to larger target shapes and higher number of available blocks (note that in all tables the * denotes evaluation on the training configuration, while the rest of the columns are out-of-distribution evaluations).
> >
> > Additionally, we are currently running new experiments with the robot to showcase the ability of the algorithm to scale up to different building blocks, and experimental settings. However, limitations still exist when it comes to introducing new building blocks that are not made up from primitive blocks (e.g., triangles, spheres, etc.). To tackle this problem, we will have to provide a hierarchical method that on a higher level of abstraction encodes the object's shape, or mesh as a graph, and on the lower level performs the reasoning that we are currently describing in the paper. We wish to research further this direction in the future.
> >
> > Regarding the transferability of the proposed method, the MHA-GNN-RL method could be useful to perform tasks of sequential pick-and-placing, as for example the table re-arrangement task. For that, an initial goal encoding of the desired object configuration should be provided (that would replace a target shape), and subsequently the unordered objects of the scene will become nodes in a graph. In that way, we could perform long-horizon planning, without needing a strict task plan definition, but the ΜΗΑ-GNN-RL+search method that we propose would be able to discover the desired goal configuration. Of course, we would require to incorporate a more complex action space, and a higher level of abstraction (e.g., first use of MHA-GNN-RL for deciding over object to move and placing position, and on a second level decision over the 6D grasp on the object to realize a successful placing).

---

> > > ### Author Response · Authors · 2021-08-27
> > > **Response to reviewer a8Bf (3/3)**
> > >
> > > >Many of the experiments are conducted without the robot — what is the motivation behind this? Perhaps the reason is that introducing the robot may introduce confounding factors in the analysis, in which case the paper should clearly justify why this may be the case
> > >
> > > The initial design ablations happened without the robot, to speed up the process of finding the optimal representation for our problem, and the most effective algorithmic solution. Including motion generation for the low-level execution requires more computational time. We believe that the addition of the robot does not provide confounding factors in the chosen architecture of the graph representation, or the generalization power of the combination of learning and planning. The robot is not encoded in the graph, but is the means for operating the geometric motion of the blocks. The use of the robot increases the action space. Hence, any algorithmic design in the ablation that could not solve the assembly task reliably with the reduced action space (lower combinatorial complexity) would not be able to resolve the robotic tasks with the increased action space. For proving the efficiency of the proposed algorithmic solution, we provided new experiments with the robot in the loop, with a more complex object set, that we will provide soon.
> > >
> > > >Numerous citations to GNN-based works are missing, including but not limited to (1) Relational deep reinforcement learning, by Zambaldi et al, (2) Relational inductive bias for physical construction in humans and machines, by Hamrick et al, (3) Contrastive learning of structured world models, by Kipf et al
> > >
> > > We thank the reviewer for pointing out these important works, which we have added in the updated related works section.

---

> ### Author Response · Authors · 2021-08-31
> **Information on final version of the paper and added experiments**
>
> We have updated our experimental results that we hope will address your concerns. Firstly, as suggested, we have added the comparison to the single-head attention mechanism in Appx. D.2.1 and in the videos on our website (https://sites.google.com/view/learn2assemble). The results show that the single attention is not enough to encode the increased relational complexity in the assembly scenes, however it still performs better than the structure2vec representation.
>
> Secondly, we have provided an evaluation of a trained agent with the robot-in-the-loop in a scene with randomized initial block positions (as compared to the structured setting during training), where we show the flexibility of the multi-head-attention graph representation to encode spatial relations without overfitting to a specific setting (see Table 4 as well as the accompanying video). This also shows that the agents can indeed generalize to novel scenes where the parts are initialized differently.
>
> Thirdly, we have provided a more complex experimental scenario where we use different objects to assemble target shapes (see Fig. 4e and Table 5). We demonstrated that the proposed MHA-GNN-RL algorithm manages to encode the spatial relations between various complex shapes and the desired target shape, as the performance of the agent trained without the robot, is the same as with the simple object-set. When training with the robot, we get a performance of up to ~ 80% success rate (94% without the robot), when evaluating on the same number of provided building blocks as with training. However,  when we add more objects the performance reduces to ~ 40% (~ 60% without the robot) with ~ 30 available blocks. Please note that in all these experiments the generation of initial objects in the scene is random, and we do not know whether a structure could actually be built effectively with the available blocks.  We attribute this performance to the limited action space of the robot (we only control planar orientations), and the fact that certain placements would need a fine insertion policy rather than a “rough” placing that leads to unstable structures. We also think that adding 6D grasps would improve the performance in these more complicated settings. We have also added an appropriate discussion in the updated version of the paper at the Remarks paragraph at the end of Section 3. We conclude from these experiments that the learned representations can indeed also generalize in scenarios with different available building blocks but do note that the current definition of the action space in combination with the low-level controller limits the method’s performance in the scenarios with the more complex objects and including the robot. Overcoming these limitations will be subject to future work.

---

> > ### Comment · Reviewer_a8Bf · 2021-09-02
> > **reviewer response to author rebuttal**
> >
> > I would like to thank the authors for thoroughly responding to my review and running additional experiments. After a careful study of the rebuttal and the revised paper, I have decided to maintain my recommendation at weak reject. While I really appreciated the effort that the authors put into the rebuttal, I am still concerned about a few matters:
> > * planning seems to be an integral part of this method. Unfortunately, it does not seem feasible to perform planning when deploying this method in the real world (especially at test time), unless a learned model is available to hallucinate future outcomes. Since a learned model was not explored in this work, the scalability of the method at the current state appears limited.
> > * the generalization experiments are quite helpful, but I was hoping they would include more complex shapes (rather than a composition of the existing primitive shapes).
> > * (more minor point) I would have preferred if the policy was trained on a larger number of objects, different shapes, different initial positions etc, rather than just *tested* on these configurations. If beleive this was not the case from reading the paper, but if it was the case, the paper should make this clear.

---

> > > ### Author Response · Authors · 2021-09-02
> > > **Response to reviewer**
> > >
> > > We thank you for looking really carefully at our revised version of the paper.
> > >
> > > We do agree that investigating different object shapes besides the concatenation of primitive boxes would indeed be very interesting - however, adding this extension was unfortunately impossible due to the lack of time, but will definitely be investigated in future work.
> > >
> > > We are also sorry to have not made it more clear that we indeed trained novel agents for the more complicated objects.
> > >
> > > Regarding the planning process when facing a real-world scenario, our method can easily be extended to address the reviewer's concerns. In the real setting, we would still make use of the simulated environment, which will be set to exactly mirror the state of the real scene. Using the simulation as a so-called digital twin and exploiting it for search or offline planning is a common practice in robotics and has actually been shown successful in many works such as the recent work  https://arxiv.org/pdf/2101.02842.pdf. This would allow us to perform the search procedure with the small budgets as proposed in the current version of the paper. We are aware that greater search budgets (magnitude of 100’s or 1000’s) are infeasible when relying on the simulator and not having a learned model available, however, with the budgets that we propose and investigate in the paper (5-10) this is an absolutely feasible task, also in the light of dealing with a real-world setup. Also, since the execution of the stacking action will for sure take around 10-20 seconds on the real system, waiting for like 2-3 seconds to conduct the search in simulation should be acceptable.
> > >
> > > We further hope to provide a real-world toy experiment for the camera ready version of the paper.

---

> > > > ### Comment · Reviewer_a8Bf · 2021-09-03
> > > > **regarding real-world experiments**
> > > >
> > > > I would be happy to reconsider my recommendation if there is a real-world experiment demonstrating planning with a digital twin. The sim2real gap makes this a non-trivial challenge.

---

### Official Review · Reviewer_7iDb · 2021-07-23

**Originality:** Very Good
**Technical Quality:** Very Good
**Clarity Of Presentation:** Very Good
**Impact:** 4

**Recommendation:**

Weak Accept: I recommend accepting the paper, but will not argue for my recommendation if the majority of other reviewers have a different opinion.

**Summary:**

This paper addresses the problem of autonomous assembly. Goals are defined by a set of 3D points forming a polygon shape that might be 2D or 3D. The robot's actions then consist of picking and placing blocks to construct the structure defined by the shape formed from the 3D points. The task notably has challenging characteristics like combinatorial explosion from a growing number of action, as well as others. A GNN representation with multi-headed attention mechanisms is proposed to represent the state of the constructed blocks and several RL methods are explored to perform model-based and model-free training. The work compares to the prior work structure2vector in one experiment, showing that the proposed architecture has more representational power essential to abstract robot assembly.

**Issues:**

Would it be possible to write Section 2.2 to rely less on existing algorithms from prior work? This might help readers that are unfamiliar with those works.

**Reviewer Expertise:**

Fair: Some knowledge of the area

**Strengths And Weaknesses:**

Strengths:
-The paper is clear, concise, and well written.
-The benchmarking environment presents another interesting problems for researchers.
-The approach presents an interesting formulation of techniques to improve robot assembly from scratch

Weakness:
-The paper differentiated itself often in the related work that had a predefined goal in the form of a demonstration, but also has a predefined goal in the form of the 3D points. Perhaps some emphasis should be place on the abstraction level of the goal.

Side comments:
-Interesting future direction I thought would define the goals in natural language.

**Summary Of Recommendation:**

I am unfamiliar with existing works on the robot structure assembly problem although I am very familiar with GNNs that include multi-headed attention mechanisms. As a result, I am assuming the main contributions applying the GNN with multi-headed attention mechanism to the robot assembly problem, robot assembly benchmark, and different combinations of learning algorithms for the problem. If other reviewers had strong opinions about missing baselines or prior work I would be open to those discussions. This is the reason for weak accept instead of strong.

---

> ### Author Response · Authors · 2021-08-27
> **Response to reviewer 7iDb**
>
> We thank the reviewer for the positive review and for pointing out unclear parts of our paper. We have updated our paper and the appendix to clarify better the addressed weaknesses and issues. Moreover, we are currently running and evaluating additional experiments, as requested by other reviewers, to showcase the generalization abilities of the proposed method (we will add those in a follow-up update of the paper).
>
> >Weakness: -The paper differentiated itself often in the related work that had a predefined goal in the form of a demonstration, but also has a predefined goal in the form of the 3D points. Perhaps some emphasis should be place on the abstraction level of the goal.
>
> We have updated the related works section to better clarify the differentiation of our work compared to the state-of-the-art. To clarify this point, all prior works know a priori, either through hard-coding or through demonstrations, the exact goal positions of every block. In our case, we have a target shape (a design) that we need to fulfill. Our problem can be viewed as that of an area covering problem. The algorithm does not know a priori the order and the positions of the blocks, and it might even well be that not all blocks are needed to construct the structure. For example, if we have to only build a tower, the algorithm has to discover that it has to place the blocks on top of each other, while previous works know the order and target position of the blocks, that leads to learning policies that learn to place (which in essence can be resolved by planning if you know already an exact goal position).
>
> We attribute our advancement in 3D robot assembly to the power of the multi-head graph attention network that can provide geometric reasoning regarding the shape covering task. As the problem of assembly is a combinatorially hard problem (most robotics algorithms solve simple structures with up to 9 blocks), we investigated different algorithmic combinations and network designs, which coupled with reinforcement learning, search and low-level motion generation with the robotic manipulator, provide an algorithm that can perform abstract reasoning on the high level, and informed assembly execution on the low-level (the robot learns to place the objects in such an order as to avoid destroying the already built structure) in one single pipeline.
>
> >Side comments: -Interesting future direction I thought would define the goals in natural language.
>
> Regarding specifying goals in natural language, we agree with the reviewer that it would be an interesting research direction for long-horizon manipulation, e.g., instruct in natural language to rearrange objects on a table, or to assemble a structure whose target shape is known (like a house), and embed semantic information to the goal nodes, or to the actions, and use MHA-GNN-RL+search method to discover the goal poses of the objects on a geometric level.
>
> >As a result, I am assuming the main contributions applying the GNN with multi-headed attention mechanism to the robot assembly problem, robot assembly benchmark, and different combinations of learning algorithms for the problem.
>
> We have updated the paper to make our contributions clearer.
>
> >Would it be possible to write Section 2.2 to rely less on existing algorithms from prior work? This might help readers that are unfamiliar with those works.
>
> We have added the complete description of all the algorithms in Section 2.2. in the supplementary material (cf. Appendix B) and added the respective reference to the updated version of the paper. We hope that this helps the readers to follow the explanations in the paper.

---

> > ### Comment · Reviewer_7iDb · 2021-09-03
> > **reviewer response to author rebuttal**
> >
> > I would like to thank the authors for their detailed response and revisions. The authors have addressed the clarifications I mentioned. My score remains the same.

---

> > > ### Author Response · Authors · 2021-09-03
> > > **Thank you**
> > >
> > > Thank you for acknowledging our efforts.

---

> ### Author Response · Authors · 2021-08-31
> **Information on final version of the paper and added experiments**
>
> We have updated our experiments, enclosing new results regarding the performance of the trained agent in randomized scenes, in which the performance remains almost intact (see Table 4) These results underline the generalization properties and the expressivity of the proposed graph-based multi-head attention representation (MHA-GNN). We have also conducted experiments with a more complex set of objects. On the one hand, the MHA-GNN-RL algorithm manages to encode the spatial relations between various complex shapes and the desired target shape, as the performance of the agent trained without the robot is the same as with the simple object-set. On the other hand, while training with the robot can deliver up to ~ 80% success rate (94% without the robot) when evaluating on the same number of provided building blocks as with training, dealing with more objects results in the performance decreasing to ~ 40% (~ 60% without the robot) with ~ 30 blocks available. However, we should note that in all these experiments the generation of initial objects in the scene is random, and we thus do not know whether a structure could actually be built effectively with the available blocks.
>
> One of the limitations that is also discussed in the updated version of the paper, is the limited number of placement actions. The problem is that certain desired placements can only be realized when controlling the 3D orientation of the end effector, which is however not implemented in the current setup (we only control the planar orientation of the end-effector around the z-axis). Also, many actions require a fine insertion that would be very difficult to execute with planning, meaning that in such cases the underlying low-level controller should rather be learned. Finally, we also believe that allowing more complex grasps in 6D would alleviate certain collisions with other blocks, and for that we believe that a further encoding of the building blocks (e.g., as a mesh-graphs) would allow us to encode relational grasp poses w.r.t. the assembled structure.

---

### Official Review · Reviewer_o7Zw · 2021-07-23

**Originality:** Fair
**Technical Quality:** Fair
**Clarity Of Presentation:** Good
**Impact:** 3

**Recommendation:**

Weak Accept: I recommend accepting the paper, but will not argue for my recommendation if the majority of other reviewers have a different opinion.

**Summary:**

This work proposes a pipeline to train a robot to learn to assemble blocks to match the given target shape. A graph neural network is utilized to encode the state information, and a deep reinforcement learning algorithm is applied to maximize the reward. The authors conduct extensive experiments to demonstrate the effectiveness of the proposed approach.

**Issues:**

1) in Line 102, "desired shape that needs to be assembled consists of a set of points parameterizing this target shape". It's unclear how the target shape is parameterized by the set of points.
2) In Line 111, "we have to decide over Na = 2 × 2 × 5 = 20 actions per block." It's difficult for the reviewer to understand these 20 actions per block. It would strengthen the paper if a visualization image is introduced here.
3) In Line115-115, what is the function of F in equation (1).
4) In Line 118-119, "a threshold ∆ is exceeded, or whenever there are no more unplaced blocks available, or after an invalid action. " What is an invalid action?


**Reviewer Expertise:**

Good: General knowledge of the area

**Strengths And Weaknesses:**

Strengths
1) This work proposes to leverage a graph neural network to extract the state information used to learn an assembly policy in a reinforcement learning fashion.
2) Extensive evaluation experiments are conducted to demonstrate the effectiveness of the proposed approach.

Weaknesses:
1) The technical contributions are limited. Using graph neural networks to represent the state and deal with the combinatorial problem in the assembly process has been investigated by several works. For example, "Generative 3D Part Assembly via Dynamic Graph Learning. Huang et al. Neurips 2022."
2) Some important descriptions are missing. For example, it's not clear why a set of points is sufficient to represent the target shape. How the topology information is encoded in the set of points?
3)  The current setting is too simple. It's not clear what's the performance if more complicated objects are used. Whether the proposed pipeline could deal with more complicated objects? How to define the action space since there are more grasping poses?



**Summary Of Recommendation:**

This work proposes a pipeline to train a robot to learn to assemble blocks to match the given target shape leveraging the graph neural network to encode the state. However, the technical novelty is limited and the current setting is too simple. It's unclear whether the proposed pipeline could work with complicated objects.

---

> ### Author Response · Authors · 2021-08-27
> **Response to reviewer o7Zw**
>
> We thank the reviewer for the thoughtful review. We hereby seek to address most of your concerns, while we are conducting additional experiments on more complex environments with more complicated (tetris-like blocks) and randomized block positions.
>
> >The technical contributions are limited. Using graph neural networks to represent the state and deal with the combinatorial problem in the assembly process has been investigated by several works. For example, "Generative 3D Part Assembly via Dynamic Graph Learning. Huang et al. Neurips 2022.
>
> Regarding the technical contribution, we propose an algorithmic solution for learning long-horizon manipulation for robotic assembly with increased action spaces. We believe that a graph neural network approach seems a reasonable solution for representing structures, as also proposed in the paper that the reviewer suggested (we thank you for pointing us at this reference, and added it to our list of related works). However, we would like to point out the combinatorial complexity of our problem compared to the aforementioned paper. The number of modules used is limited, but we acknowledge the fact that the related method reasons on a 6D action space. A more elaborate design of the graph neural network is necessary to be able to encode the relations between an increasing number of objects, and to encode the actions in a reinforcement learning setting that will allow to reason about picking actions and placing positions and orientations. In the future, we are interested in incorporating 6D object pose detection and 6D grasping in our framework, but that would need a higher level of abstraction of the action space that is not in the scope of the current paper.
>
> >Some important descriptions are missing. For example, it's not clear why a set of points is sufficient to represent the target shape. How the topology information is encoded in the set of points?
>
> Regarding the topology of the target shape, this is encoded by the nodes and edges in the graph that represent the shape that needs to be fulfilled. All the building blocks are connected via edges to the target nodes, as to reason about their spatial connectivity. We have updated the paper in the description of the goal shape, to make this clearer to the reader (see updated introduction in Sec. 2 and Fig. 2).
>
> >The current setting is too simple. It's not clear what's the performance if more complicated objects are used. Whether the proposed pipeline could deal with more complicated objects? How to define the action space since there are more grasping poses?
>
> Our proposed algorithm can handle different objects that can be viewed as a combination of primitive shapes. This can be seen in the experiment of Fig. 4e and in the updated video, where we have tested the combinatorial abilities of the proposed algorithmic solution when combining different object shapes.
> To showcase the efficiency of the proposed graph structure to solve more challenging tasks with the robot in the loop, we are conducting new experiments with different objects. We will update the paper as soon as our full evaluation is finalized. Initial findings show generalization to different object sets, but also give insights on the limitations and possible solutions that we will discuss accordingly.
>
> We need to acknowledge that all algorithmic solutions suffer from a combinatorial burden. We believe that a future direction would be to also encode the building blocks as graphs. This would allow us to reason on a continuous action space. However, in continuous action spaces the combinatorics are infinite, making this a hardly tractable problem. This is why we choose to discretize the action space (common practice in robot research) and allow for the continuous execution of the assembly with a robot (touching on the ideas of task and motion planning).
>
> Regarding the issues:
> 1. We clarified the description of the target shape (see introduction of Sec. 2 and Fig. 2).
> 2. We have significantly rephrased the description of the action space (in the introduction of Sec. 2) and added a visualization of the grasping and placing poses in Appendix C.6 Fig. 2.
> 3. We are sorry for the poor description of the reward function. While we updated Sec. 2 as to “We assign a positive reward $r(s_t,a_t)$ on all actions that increase the filling of the target structure while preserving its stability.”, we have also included a thorough description of the reward function in Appendix C.2, due to the limited amount of space.
> 4. An invalid action is an action that disrupts the assembly (e.g., the robot crashing into the built structure or a kinematically infeasible action). For the experiments without the robot, every action that results in an unstable configuration of the placed elements is invalid. As described in the updated version of the paper in Appendix C.2, as well as in Section 3, every invalid action leads to a negative reward signal and results in resetting the environment.

---

> ### Author Response · Authors · 2021-08-31
> **Information on final version of the paper and added experiments**
>
> As we have promised, we have conducted further experiments and updated our results in the paper and the appendix. The more complicated object set is visualized in Fig. 4e and more information can be also found in Appx. D.9. The complex objects are composed of primitive blocks (boxes). This choice allows us to keep the relative action space, i.e., placing an unplaced primitive box belonging to a larger object w.r.t. a placed primitive box results in moving the entire object. The results from running the additional experiments can be inspected in Table 5.
>
> While the proposed MHA-GNN-RL method can handle the increased combinatorial complexity of the more complex objects, when training with the robot-in-the-loop, we experience a drop in performance compared with the simple scenarios (though the structures and combinatorics are much higher), due to limitations of our motion generation strategy (e.g., the robot might move a piece that might collide with the structure, or places a block that will not be stable and can fall).
>
> On the one hand, the MHA-GNN-RL algorithm manages to encode the spatial relations between various complex shapes and the desired target shape, as the performance of the agent trained without the robot is the same as with the simple object-set. On the other hand, while training with the robot can deliver up to ~80% success rate (94% without the robot) when evaluating in the training environments, adding more objects results in reduced performance with ~40% success (60% without the robot) with ~30 available blocks. However, we should note that in all these experiments the generation of initial objects in the scene is random, and we do not know whether the sampled structure can actually be built with the available blocks.
>
> One of the limitations that is also discussed in the updated version of the paper, is the limited number of placement actions. The problem is that certain desired placements can only be realized when controlling the 3D orientation of the end effector, which is however not implemented in the current setup (we only control the planar orientation of the end-effector around the z-axis). Also, many actions require a fine insertion that would be very difficult to execute with planning, meaning that in such cases the underlying low-level controller should rather be learned. Finally, we also believe that allowing more complex grasps in 6D would alleviate certain collisions with other blocks, and for that we believe that a further encoding of the building blocks (e.g., as a mesh-graphs) would allow us to encode relational grasp poses w.r.t. the assembled structure.

---

### Author Response · Authors · 2021-08-27
**General comment**

We thank all the reviewers and the metareviewer for their thoughtful evaluation of our work. We are currently working to substantially update the paper and add more experimental results (i.e., comparing single-head attention to multi-head attention; evaluating our policies with more randomized initial block positions, and variable building blocks).

In the following, we provide replies to your comments, as to initiate the discussion. We have also updated the paper and the supplementary material, with updated Appendix, and the diff.pdf of the current version of the paper with the changes marked in blue.

Please note that we will update once more our responses and the paper, as soon as our additional experiments are completed.

---

### Author Response · Authors · 2021-08-31
**Updated Results**

As we promised earlier, we have completed additional experiments, and have now updated the paper and the supplementary material accordingly. In the supplementary material, you will find an updated appendix, the diff.pdf file which highlights the changes of the updated version compared to the initial submission (changes marked in blue), and a new video showcasing additional results from the new experiments.

Note that while preparing the updated version, we discovered a small bug in the robotic experiments (building of the 2- and 4-sided structures) and thus updated the results of Tables 3 and 4 accordingly (note that in the old version of the paper, these results were reported in Tables 5 and 6).

---

### Meta-Review · Area_Chair_rSR9 · 2021-08-16

**Recommendation:** Accept (Poster)
**Confidence:** 4

**Metareview:**

# Strengths
- The novelty of leveraging a graph neural network with multi-headed attention mechanism in manipulation assembly setting was noted by multiple reviewers
- Extensive evaluation experiments to demonstrate the effectiveness of the proposed approach.

# Weaknesses
- There are mixed reviews about writing clarity
- Multiple reviewers raised concerns about the simplicity of the task settings and connections to real world problems.
- Multiple reviewers noted that the technical contributions are limited
- Multiple reviewers highlight missing experimental and technical details.


# Recommendations
- Please outline limitations of the work
- Thoroughly go through the recommendations provided by each reviewers to revise the paper
- This is a borderline paper with mixed reviews, please converse and address reviewers concerns

# Decision summary
- Extensive clarifications and additional experiments were received from the authors during the rebuttal period and were considered by the reviewers before final recommendation was made.
- Authors clarifications were acknowledged and factored in. Some concern around the experiments still remains.

We hope that the feedback received will help with the future directions of the work under submission. Thank you for participating in CoRL

---

> ### Author Response · Authors · 2021-08-27
> **Response to metareview**
>
> Thank you very much for the comprehensive feedback to our paper. We have improved the writing of the paper according to your and reviewers’ recommendations, and we have added several details to the Appendix (with appropriate references to the main paper). We believe we clarified the main contributions, that are
> 1. the design of a multi-head attention graph representation for robotic assembly tasks of combinatorial complexity with different sets of building blocks and increasing number of objects,
> 2. a learning framework that combines learning and planning with the robot-in-the-loop, thus, integrating effectively the high-level action decision-making with the low-level action execution by the robot, to ensure the action feasibility and stability of the built structure, and
> 3. a benchmarking environment with modular design, that we will make public to the community upon acceptance.
>
> To answer the concerns over the task simplicity, we are currently running new experiments and evaluating agents that are able to perform assembly with the robot on complex object-sets (tetris-like modules). Our preliminary findings showcase generalization to different object sets, but also provide new insights on limitations, for which we will discuss possible future solutions.
> We will update these results on a follow-up update of the paper. However, we would like to point out that to the best of our knowledge, there are no other works on autonomous robotic assembly, without prescribed or demonstrated task plan, that can build structures of more than 10 box-shaped (simple) modules with the robot in the loop.

---

> ### Author Response · Authors · 2021-08-31
> **Information on final version of the paper and added experiments**
>
> We have now included further experimental results regarding the generalization properties and the expressivity of the proposed graph-based multi-head attention representation (MHA-GNN) when transferring to randomized scenes and when dealing with multiple different building blocks. The results show that the proposed representation is very effective for encoding spatial relations. Hence, transferring an agent trained on a “structured” scene to a more realistic one with the blocks spread out randomly, does not affect the performance significantly (see Table 4 in the updated version).
>
> Regarding the performance in more challenging environments, we are reporting both results with and without the robot. When training without the robot, we also incorporate the decisions over rotations. We can see that despite the increased action space and an increasing number of blocks, the performance remains comparable to the simple setting when only using simple boxes. When training with the robot, we have experienced a performance drop, which is attributed to the limited action space of the robot (we only control planar orientations), and the fact that certain placements would need a fine insertion policy rather than a “rough” placing that leads to unstable structures. Still, our results are promising and provide an interesting direction for future research for improving the low-level robot controller to allow us to do successful high-level reasoning with our proposed MHA-GNN-RL algorithm. We have also added an appropriate discussion in the updated version of the paper at the Remarks paragraph at the end of Section 3.
>
> We hope that this final version of the paper addresses most of the reviewers' concerns.

---

### Decision · Program_Chairs · 2021-09-13

**Decision:**

Accept (Poster)

**Comment:**

# Strengths
- The novelty of leveraging a graph neural network with multi-headed attention mechanism in manipulation assembly setting was noted by multiple reviewers
- Extensive evaluation experiments to demonstrate the effectiveness of the proposed approach.

# Weaknesses
- There are mixed reviews about writing clarity
- Multiple reviewers raised concerns about the simplicity of the task settings and connections to real world problems.
- Multiple reviewers noted that the technical contributions are limited
- Multiple reviewers highlight missing experimental and technical details.


# Recommendations
- Please outline limitations of the work
- Thoroughly go through the recommendations provided by each reviewers to revise the paper
- This is a borderline paper with mixed reviews, please converse and address reviewers concerns

# Decision summary
- Extensive clarifications and additional experiments were received from the authors during the rebuttal period and were considered by the reviewers before final recommendation was made.
- Authors clarifications were acknowledged and factored in. Some concern around the experiments still remains.

We hope that the feedback received will help with the future directions of the work under submission. Thank you for participating in CoRL